# Preserving Angles Improves Feature Distillation

**Evelyn J. Mannix**  *evelyn.mannix@unimelb.edu.au*
*School of Mathematics and Statistics*
*University of Melbourne*

**Liam Hodgkinson**  *lhodgkinson@unimelb.edu.au*
*School of Mathematics and Statistics*
*University of Melbourne*

**Howard Bondell**  *howard.bondell@unimelb.edu.au*
*School of Mathematics and Statistics*
*University of Melbourne*

**Reviewed on OpenReview:** *https://openreview.net/forum?id=ZEhgODZkWU*

## Abstract

Knowledge distillation methods compress models by training a student network using the classification outputs of a high quality teacher model, but can fail to effectively transfer the properties of computer vision foundation models from the teacher to the student. While it has been recently shown that feature distillation—where a teacher model's output features are replicated instead—can reproduce performance for foundation models across numerous downstream tasks, they fall short in matching critical properties such as robustness and out-of-distribution (OOD) detection performance. This paper overcomes this shortcoming by introducing Cosine-similarity Preserving Compression (CosPress), a feature distillation technique that learns a mapping to compress the latent space of the teacher model into the smaller latent space of the student, by preserving the cosine similarities between image embeddings. This enables direct optimisation of the student network and produces a more faithful reproduction of the teacher's properties. It is shown that distillation with CosPress on a variety of datasets, including ImageNet, produces more accurate models with greater performance on generalisability, robustness and OOD detection benchmarks, and that this technique provides a competitive pathway for training highly performant lightweight models on small datasets. Code is available at github.com/emannix/cospress.

## 1 Introduction

Deep learning computer vision approaches have become the standard for automating vision problems across a range of fields, from medical imaging (Zhang & Metaxas, 2024) to analysis of satellite imagery (Bastani et al., 2023) and detecting weapons in luggage (Andriyanov, 2024). However, models trained with commonly used supervised learning approaches can have poor robustness (Bai et al., 2021; Hendrycks et al., 2021b) and struggle to detect out-of-distribution (OOD) data (Yang et al., 2022a; Nguyen et al., 2015).

By leveraging large Vision Transformer (ViT) architectures and pretraining on large and diverse datasets, foundation models in computer vision comprise a significant step forward toward addressing these challenges, providing significantly improved generalisation ability and robustness in comparison to purely supervised approaches (Oquab et al., 2024; Radford et al., 2021). Large ViT models enjoy superior performance after pre-training (Zhai et al., 2022), and can be distilled to produce smaller models that are more practical for deployment. For example, smaller DINOv2 foundation models were distilled from their largest variant with 1.1 billion parameters by optimising the self-supervised training objective with a frozen teacher (Oquab et al., 2024). This approach is not replicable, as it was conducted on the proprietary LVD-142M dataset and the teacher head weights were never publicly released.

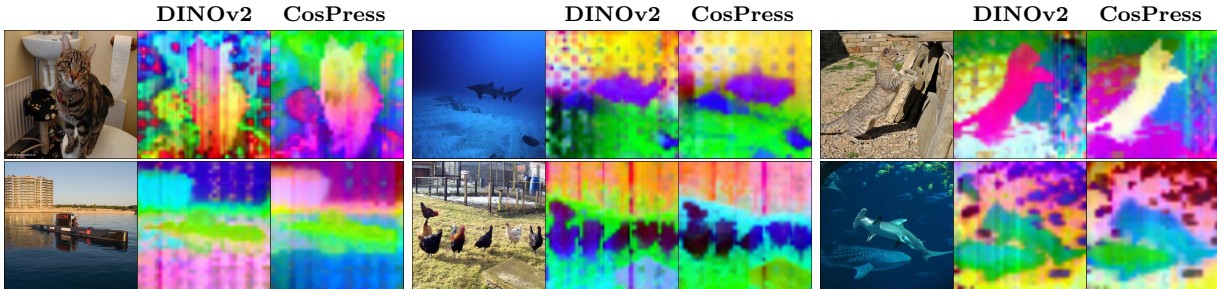

Figure 1: **Patch features.** PCA visualisation of patch features for the DINOv2 ViT-S/14 model, and the distilled ViT-Ti/14 model produced using the CosPress feature distillation approach.

Nevertheless, knowledge distillation approaches (Hinton et al., 2015) that leverage the classification outputs of the DINOv2 models trained on a particular datasets can be used to train performant student models. However, it has been shown that fundamental properties of the foundation model need not transfer to the student models, impacting generalisation performance on downstream tasks (Zhang et al., 2025). Feature distillation approaches that use a Mean Squared Error (MSE) or $L_2$ loss and a student head to map the activations from the latent space of the student to the teacher are more effective in this respect. The recent Proteus (Zhang et al., 2025) approach has shown that it is possible to distill the DINOv2 models on ImageNet-1K and obtain comparable performance on downstream classification and segmentation tasks. However, sub-optimal results are still obtained for key robustness metrics, as well as generalisability and performance in dense tasks such as image segmentation. Most concerning, however, is that models distilled using Proteus *do not faithfully reproduce* the latent space of the teacher model, as shown by their severely reduced performance on out-of-distribution (OOD) detection tasks (Table 1).

This paper presents Cosine-similarity Preserving Compression (CosPress), a feature distillation approach that addresses such shortcomings by learning a teacher head mapping that compresses the latent space of the teacher model into the latent space of the student. CosPress achieves this by preserving cosine similarities between points in the teacher's latent space and allows the student to be directly optimised. This significantly improves the faithfulness of the learned student on OOD detection (Table 1) and robustness benchmarks in comparison to Proteus, while achieving competitive performance across all of the considered challenges— including classification accuracy, generalisation and semantic segmentation. CosPress reproduces the high-quality patch features of the foundation model (Fig. 1), and can be used to produce specialised models, that have improved accuracy on particular tasks but retain these foundation model properties. In this work, we:

- present CosPress, an approach for distilling ViT foundation models that learns a mapping from the latent space of the teacher to the student that preserves cosine similarities and allows direct optimisation of the student model;

- demonstrate that CosPress *produces a more faithful student model*, better replicating the performance of the teacher across a range of metrics including robustness, generalisability and out-of-distribution detection; and

Table 1: **Out-of-distribution detection.** Comparison of performance on the OpenOOD benchmark for the ImageNet-1K dataset. The ↑ means larger values are better and the ↓ means smaller values are better.

| Method | Arch | Teacher | Near OOD | | Far OOD | |
|--------|------|---------|----------|------|---------|------|
| | | *DINOv2* | AUROC↑ | FPR↓ | AUROC↑ | FPR↓ |
| Proteus | ViT-Ti/14 | ViT-S/14 | 64.17 | 85.73 | 74.22 | 67.97 |
| CosPress | ViT-Ti/14 | ViT-S/14 | **70.49** | **77.29** | **91.03** | **37.21** |
| | | ViT-S/14 | 72.58 | 74.12 | 92.67 | 29.55 |
| Proteus | ViT-S/14 | ViT-B/14 | 61.19 | 94.56 | 61.92 | 86.78 |
| CosPress | ViT-S/14 | ViT-B/14 | **73.5** | **73.84** | **92.93** | **28.98** |

- show that CosPress *can be used to train specialised models* with improved performance on a particular vision task, while retaining foundation model properties such as improved generalisability and out-of-distribution detection performance.

## 2 Related Work

**Foundation models**  Foundation models in computer vision follow the success of transformer-based foundation models in language, such as BERT (Devlin et al., 2019), and encode images as vectors in latent space, where the distance between vectors describes the semantic similarity of the images. Two approaches have emerged for training these models: self-supervised learning (Oquab et al., 2024) and contrastive language-image pretraining (CLIP) (Radford et al., 2021). The CLIP models were among the first to show that by using a large Vision Transformer (ViT) architecture (Dosovitskiy et al., 2021), and a large, diverse and high quality training dataset, a generalist vision model could be produced that achieves high performance across a range of applications (Radford et al., 2021). The DINOv2 foundation models followed, and using a combined bootstrapping (Grill et al., 2020; Caron et al., 2021) and masked patch prediction (Zhou et al., 2022) approach to train foundation models with strong performance on image classification and segmentation tasks (Oquab et al., 2024).

**Knowledge distillation.**  Knowledge distillation is the process of transferring knowledge from a large model or model ensemble to a single smaller model. The earliest approaches aligned the output probability vectors of the student and teacher classifications using a Kullback–Leibler (KL) divergence loss (Hinton et al., 2015). There is a wide range of literature demonstrating how this approach can improve the performance of smaller Convolutional Neural Networks (CNNs) (Wei et al., 2020) and Vision Transformers (ViTs) (Touvron et al., 2021; Yang et al., 2024), by leveraging a strong teacher or one with different inductive biases to the student model. It has been shown that knowledge distillation is most effective when it is treated as a function matching problem (Beyer et al., 2022), with the same inputs being provided to both the teacher and student model.

**Feature distillation.**  Feature distillation—where the output features of the teacher are used for training the student instead of the classification outputs—is less well studied for models without class outputs. Generally speaking, feature distillation is used in combination with a knowledge distillation objective and often focuses on supervised models. However, the Proteus approach (Zhang et al., 2025) demonstrated that using pure feature distillation objectives is important for preserving foundation model properties. The components of Proteus—a student head to align output dimensions, MSE loss on class and patch tokens, and an iBOT (Zhou et al., 2022) inspired masking objective—are a logical adaption of components in prior supervised feature distillation methods such as Masked Generative Distillation (MGD) (Yang et al., 2022b), SRD (Miles & Mikolajczyk, 2024) and $V_k D$ (Miles et al., 2024) to a ViT architecture with the aim of preserving both the local and global features of the teacher.

**Dimensionality reduction.**  Feature distillation and the challenge of compressing latent spaces to train performant student models are closely related to the broader ideas of dimensionality reduction and minimum distortion embeddings (Agrawal et al., 2021). Stochastic Neighbor Embedding (SNE) (Hinton & Roweis, 2002; Van der Maaten & Hinton, 2008) is a dimensionality reduction technique that projects high-dimensional embeddings into a low-dimensional space (typically two dimensions) while preserving local relationships. SNE acheives this by constructing a probability distribution over pairs of points in the original space and then optimising a corresponding set of points in the low-dimensional space to match this higher dimensional distribution as closely as possible. However, SNE does not learn an explicit mapping between the original and reduced spaces, only a lower dimensional representation. Other approaches, in contrast, explicitly learn projection functions, often with the goal of preserving local geometric structures such as distances or angles (Saul & Roweis, 2000; He & Niyogi, 2003; Gao et al., 2020; Fischer & Ma, 2024).

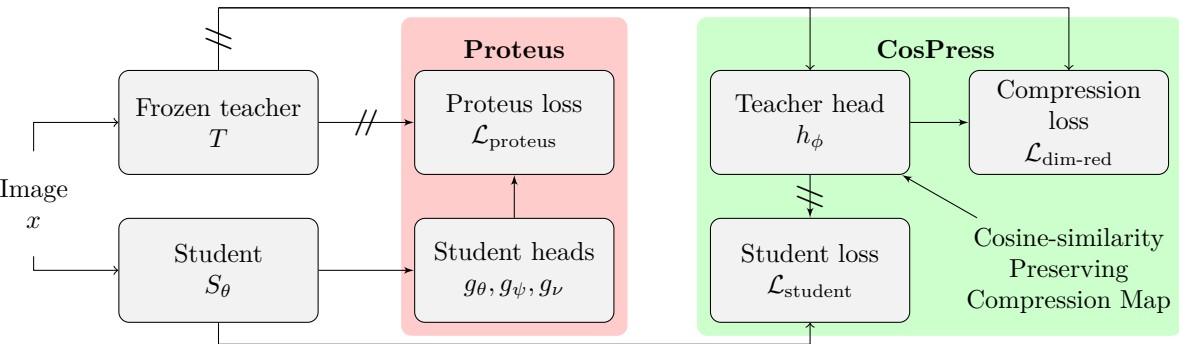

Figure 2: **Feature distillation frameworks.** In Proteus, student heads $g$ are used to map the outputs of the student network $S_\theta$ into the latent space of the teacher $T$, so that a MSE loss can be applied. In CosPress, a teacher head $h$ is trained to compress the teacher $T$ outputs into the student latent space, preserving the cosine similarity of image embeddings and allowing direct optimisation. The Proteus student head does not preserve cosine similarity, even when the projection matrices are forced to be right-orthogonal.

## 3 Methods

**Notation.** We consider a feature distillation setting, where there is a small student network $S_\theta$ with output dimensionality $D_S$ and a larger frozen teacher network $T$ with output dimensionality $D_T$. These networks use a ViT architecture, so we assume $D_T > D_S$. The loss functions presented in this paper consider a mini-batch stochastic gradient descent setting, defined for a batch of images $x_i \in \mathbf{X}$. When only the output class tokens are considered $S_\theta^c(x_i), T^c(x_i)$ is used. We write $S(x_i), T(x_i)$ to refer to a matrix of the concatenated patch and class token outputs.

**Motivation.** We are interested in the problem of training a student network to mimic the behaviour of a large, high quality teacher model using a ViT architecture, such as the DINOv2 foundation models. A key property of these models is that the cosine similarity between images and patch embeddings captures their semantic similarity, as shown by the use of this measure for zero-shot classification and identifying duplicate imagery (Jose et al., 2024; Oquab et al., 2024; Radford et al., 2021). However, larger ViT architectures have a larger output dimensionality, which prevents the embeddings produced by smaller student models being directly compared to a teacher model embedding.

Proteus (Zhang et al., 2025) addresses this problem by introducing a student head $g : \mathbb{R}^{D_S} \to \mathbb{R}^{D_T}$ that maps the outputs of the student model into the latent space of the teacher model, allowing a MSE ($L_2$) loss to be applied. This student head contains a projection matrix, $\mathbf{W} \in \mathbb{R}^{D_S \times D_T}$, that maps the from the latent space of the student to the teacher, and is *commonly discarded*. Two issues arise with this approach. First, the projection may encode information specific to replicating the teacher network, potentially distorting the outputs of the student model, which is only indirectly optimised (Miles et al., 2024). Second, there is no guarantee that the projection matrix $\mathbf{W}$ will be faithful in preserving cosine similarities between teacher embeddings of different images—similarities that reflect semantic relationships (Jose et al., 2024)—within the student's latent space.

Prior work has found that requiring $\mathbf{W}$ to be right-orthogonal addresses the first problem (Miles et al., 2024). Right-orthogonality means that $\mathbf{W}\mathbf{W}^\top = \mathbf{I}_{D_S}$ must be satisfied where $\mathbf{I}_{D_S} \in \mathbb{R}^{D_S \times D_S}$ is the identity matrix of rank $D_S$. However, this implies that for any image $i$, we have

$$S_\theta^c(x_i)\mathbf{W} \approx T^c(x_i) \implies S_\theta^c(x_i) \approx T^c(x_i)\mathbf{W}^\top. \tag{1}$$

Consequently, the cosine distance between image embeddings in the student network relates to the teacher, for any two images $i, j$, via

$$\frac{S_\theta^c(x_i) \cdot S_\theta^c(x_j)}{\|S_\theta^c(x_i)\| \, \|S_\theta^c(x_j)\|} \approx \frac{\left(T^c(x_i)\mathbf{W}^\top\right) \cdot \left(\mathbf{W}\, T^c(x_j)^\top\right)}{\|T^c(x_i)\mathbf{W}^\top\| \, \|\mathbf{W}\, T^c(x_j)^\top\|}. \tag{2}$$

This implies that $\mathbf{W}$ also needs to be left-orthogonal to address the second problem and ensure that cosine similarities are preserved. Asserting left-orthogonality with $\mathbf{W}^\top\mathbf{W} = \alpha\mathbf{I}_{D_T}$ for some scalar $\alpha$ would yield the desired relationship

$$\frac{S_\theta^c(x_i) \cdot S_\theta^c(x_j)}{\|S_\theta^c(x_i)\| \, \|S_\theta^c(x_j)\|} \approx \frac{T^c(x_i) \cdot T^c(x_j)^\top}{\|T^c(x_i)\| \, \|T^c(x_j)^\top\|}. \tag{3}$$

Unfortunately, the projection $\mathbf{W}$ can only be *approximately* left-orthogonal, as $\mathbf{W}$ is only of rank $D_S$, which means the product $\mathbf{W}^\top\mathbf{W}$ can only ever be of rank $D_S$. As $\mathbf{I}_{D_T}$ is of rank $D_T > D_S$, then $\mathbf{W}^\top\mathbf{W} \neq \alpha\mathbf{I}_{D_T}$. Consider the following definition.

**Definition 1** (Approximately Orthogonal Matrix). A matrix $\mathbf{M} \in \mathbb{R}^{m \times d}$ is said to be *approximately orthogonal* if

$$\|\mathbf{M}\mathbf{M}^\top - \alpha\mathbf{I}_m\|_F < \varepsilon \quad \text{and} \quad \|\mathbf{M}^\top\mathbf{M} - \beta\mathbf{I}_d\|_F < \varepsilon,$$

for sufficiently small $\varepsilon > 0$ and real scalars $\alpha, \beta$. If only one of these conditions are satisfied the matrix is said to be approximately right or left orthogonal, as appropriate. This notion expands upon the idea of orthogonality, where a matrix $\mathbf{M}$ can only be orthogonal ($\varepsilon = 0, \alpha = \beta = 1$) if it is square ($m = d$).

**Lemma 1.** *Let $\mathbf{M} \in \mathbb{R}^{m \times d}$ with $m < d$ and $\mathrm{rank}(\mathbf{M}) = m$. Then*

$$\left\|\mathbf{M}\mathbf{M}^\top - \tfrac{d}{m}\mathbf{I}_m\right\|_F \leq \left\|\mathbf{M}^\top\mathbf{M} - \mathbf{I}_d\right\|_F.$$

*Moreover, the converse inequality does not generally hold.*

This lemma shows that approximate right-orthogonality is sufficient for a matrix to also be approximately left-orthogonal, and therefore to be approximately orthogonal overall. Consequently, both conditions Eq. (1) and Eq. (3) are satisfied, ensuring that the mapping does not encode information while preserving the relationships between image embeddings. The proof of Lemma Theorem 1 stems from $\mathbf{M}$ having more columns than rows, which can be downsampled to form a square matrix, and is provided in Section A of the supporting information. While prior methods such as $V_k D$ introduce parametrisations that require $\mathbf{W}$ to be right-orthogonal (Miles et al., 2024), they do not also guarantee approximate left-orthogonality.

While Eq. (3) could be optimised directly using an SNE (Hinton & Roweis, 2002; Van der Maaten & Hinton, 2008) inspired approach, our initial experiments found that this was less effective than Proteus due to this target having a complex loss surface with many local minima. Instead, it is proposed to use a teacher head, rather than a student head, to learn a function $h_\phi : \mathbb{R}^{D_T} \to \mathbb{R}^{D_S}$ that compresses the representation of the teacher into the latent space of the student while preserving cosine similarities

$$\frac{T^c(x_i) \cdot T^c(x_j)}{\|T^c(x_i)\| \, \|T^c(x_j)\|} \approx \frac{h_\phi(T^c(x_i)) \cdot h_\phi(T^c(x_j))}{\|h_\phi(T^c(x_i))\| \, \|h_\phi(T^c(x_j))\|}, \tag{4}$$

which allows the student $S_\theta^c(x_i)$ to be directly optimised against the compressed teacher representation $h_\phi(T^c(x_i))$. Learning the $h_\phi$ mapping is a tractable problem as described by the Johnson–Lindenstrauss (JL) Lemma, which states that such a mapping can be constructed with a margin of error that depends on the dimensionality of the target space and the size of the dataset of interest (Freksen, 2021). Further details are provided in Section A, and Fig. 2 highlights the differences between the Proteus and CosPress frameworks.

**Proteus.** Zhang et al. (2025) propose to minimize the MSE ($L_2$) loss between the outputs of the teacher and that of the student, when passed through a dimension-raising map called the student head $g : \mathbb{R}^{D_S} \to \mathbb{R}^{D_T}$. To achieve best performance, they use three student heads with different weights $\phi, \psi, \nu$ and minimise the $L_2$ loss separately on the class tokens, features (class and patch tokens), and on randomly masked tokens $\mathbf{X}^M$ similar to the MGD (Yang et al., 2022b) approach. This leads to the following optimisation loss

$$\begin{aligned}
\mathcal{L}_{\mathrm{proteus}}(\mathbf{X}; \phi, \psi, \nu, \theta) = \; & L_2(g_\phi(S_\theta(\mathbf{X})), T(\mathbf{X})) \\
& + L_2(g_\psi(S_\theta^c(\mathbf{X})), T^c(\mathbf{X})) \\
& + L_2(g_\nu(S_\theta(\mathbf{X}^M)^M), T(\mathbf{X})^M).
\end{aligned} \tag{5}$$

**CosPress.** Our approach, CosPress, separates the challenge of feature distillation into two parts,

$$\mathcal{L}_{\text{CosPress}}(\mathbf{X}; \phi, \theta) = \mathcal{L}_{\text{dim-red}}(\mathbf{X}; \phi) + \mathcal{L}_{\text{student}}(\mathbf{X}; \theta). \tag{6}$$

Firstly, a teacher head $h_\phi : \mathbb{R}^{D_T} \to \mathbb{R}^{D_S}$ is learnt to map the teacher outputs $T$ to the latent space of the student network $S_\theta$ while preserving cosine similarities. This dimensionality reduction loss term $\mathcal{L}_{\text{dim-red}}$ is independent of fitting the student network. It only requires the target dimension in order to fit the teacher head $h_\phi$.

Secondly, the student network $S_\theta$ is trained to match the image of the teacher under the teacher head $h_\phi \circ T$ in the student loss term $\mathcal{L}_{\text{student}}$. The most effective way to fit this term is to freeze the teacher head $h_\phi$ gradients and train on both losses concurrently as shown in Fig. 2. Using a weighting scheme was observed to produce similar results (Section C).

**Dimensionality reduction objective.** To build a loss function that will ensure the mapping $h_\phi$ satisfies Eq. (4) an SNE (Hinton & Roweis, 2002; Van der Maaten & Hinton, 2008) inspired approach is used. This involves defining a kernel to build distributions describing the similarity between vectors, allowing for embeddings in the high dimensional input space to be aligned with the low dimensional target space by minimising the KL divergence between these distributions.

We define a kernel using the von-Mises Fisher distribution, where for input vectors $y$ and $z$ we have

$$k_\tau(y; z) \propto \exp\left(\frac{y \cdot z}{\|y\| \|z\|} / \tau\right), \tag{7}$$

with temperature hyperparameter $\tau$. As a result, Eq. (4) becomes

$$k_\tau\left(T^c(x_i); T^c(x_j)\right) \approx k_\tau\left(h_\phi(T^c(x_i)); h_\phi(T^c(x_j))\right), \tag{8}$$

for each $i, j$. Then, for a set of vectors in the input space $p_i \in \mathbf{p}$ and target space $q_i \in \mathbf{q}$ of size $N$, we construct the matrices $P^\tau, Q^\tau$ that define the input and target distributions by

$$P_{ij}^\tau = \frac{p_{j|i} + p_{i|j}}{2N}, \quad p_{j|i} = \frac{k_\tau(p_i; p_j)}{\sum_{i \neq k} k_\tau(p_i; p_k)}, \tag{9}$$

$$Q_{ij}^\tau = \frac{q_{j|i} + q_{i|j}}{2N}, \quad q_{j|i} = \frac{k_\tau(q_i; q_j)}{\sum_{i \neq k} k_\tau(q_i; q_k)}, \tag{10}$$

where the first equation builds symmetric $P^\tau, Q^\tau$ matrices, allowing for greater flexibility in the solution. If these $P^\tau, Q^\tau$ matrices are equal, the cosine similarity between pairs of points in $\mathbf{p}, \mathbf{q}$ will be equal and Eq. (4) will be satisfied. This can be achieved approximately by minimising the KL divergence ($D_{\text{KL}}$) over $\boldsymbol{\tau}$, a vector of temperature values via

$$L_{\text{KL}}(\mathbf{p}, \mathbf{q}) = \frac{1}{|\boldsymbol{\tau}|} \sum_{\tau \in \boldsymbol{\tau}} D_{\text{KL}}(P^\tau \| Q^\tau). \tag{11}$$

An ablation study on the best values of $\boldsymbol{\tau}$ is described in Table S21 in the supporting information.

Putting this all together, we propose a dimensionality reduction loss that conserves cosine similarity at two levels—between the image class tokens in a batch, and between the features (patch and class tokens) within an image

$$\mathcal{L}_{\text{dim-red}}(\mathbf{X}; \phi) = L_{\text{KL}}(h_\phi(T^c(\mathbf{X})), T^c(\mathbf{X})) \tag{12}$$

$$+ \frac{1}{|\mathbf{X}|} \sum_i L_{\text{KL}}(h_\phi(T(x_i)), T(x_i)).$$

The calculation of $L_{\text{KL}}(h_\phi(T^c(\mathbf{X})), T^c(\mathbf{X}))$ is the only term in the CosPress loss that is calculated between examples in a batch, and that will scale non-linearly with increasing batch size. All other terms in both CosPress and Proteus are computed within individual examples and scale linearly with batch size.

**Teacher head architecture.** As done for the Proteus (Zhang et al., 2025) student heads, the teacher head architecture in CosPress uses a LayerNorm (Ba, 2016) followed by a linear layer. This can be written as

$$h_\phi(z) = \left( \frac{z - \bar{z}}{\|z\|} \gamma + \beta_1 \right) \mathbf{W}^\top + \beta_2, \tag{13}$$

where $\bar{z}$ is the average of the $z$ vector elements, and the initialisation scheme sets the biases $\beta_1, \beta_2$ to zero and the scaling $\gamma$ to one at the start of training. The linear map $\mathbf{W}^\top \in \mathbb{R}^{D_T \times D_S}$ is initialised using a random normal distribution as is standard, which is consistent with the mapping constructed in the JL Lemma (Section A). For the Proteus student heads $g$, $\mathbf{W}^\top$ is replaced by $\mathbf{W}$ and the dimension of the bias vectors are adjusted accordingly.

**Student objective.** The student objective minimises cosine distance

$$L_{\text{cosine}}(\mathbf{z}, \mathbf{y}) = \frac{1}{n} \left( \sum_i 1 - \frac{z_i \cdot y_i}{\|z_i\| \, \|y_i\|} \right), \tag{14}$$

where $\mathbf{z}, \mathbf{y}$ are sets of input vectors of the same length $n$. Considering the teacher head $h_\phi$ learns to conserve cosine similarity, this is a natural choice for the student network and is found to result in improved performance with CosPress than the $L_2$ loss (Section C).

Similarly to the dimensionality reduction objective, the final student loss employs both a class token loss and a feature loss term

$$\begin{aligned} \mathcal{L}_{\text{student}}(\mathbf{X}; \theta) &= L_{\text{cosine}}(S_\theta^c(\mathbf{X}), h_\phi(T^c(\mathbf{X}))) \\ &\quad + L_{\text{cosine}}(S_\theta(\mathbf{X}), h_\phi(T(\mathbf{X}))). \end{aligned} \tag{15}$$

## 4 Experiments: Feature distillation

In this section, the CosPress feature distillation approach is compared to Proteus and distilled variants of the DINOv2 models. While we do not have access to the proprietary LVD-142M dataset used to distill the DINOv2 models, it has been shown that ImageNet-1K (Russakovsky et al., 2015) is sufficient to distill models with comparable accuracy across a range of measures (Zhang et al., 2025).

### 4.1 Experimental setup

Vision Transformer (Dosovitskiy et al., 2021) models are distilled using larger DINOv2 teachers on the ImageNet-1K (Russakovsky et al., 2015) training dataset, comprising 1000 categories across more than 1.2 million training images. To enable a fair comparison, we reproduce the results of the Proteus paper and train CosPress models using a unified codebase. This ensures consistency in the optimizers, samplers, augmentations and other hyperparameters. Following Proteus (Zhang et al., 2025), student networks are distilled for 300 epochs using a batch size of 1024, cosine learning rate decay with five warmup epochs (Loshchilov & Hutter, 2017a), an AdamW optimizer (Loshchilov & Hutter, 2017b), a repeated augmentation sampler with three views per image (Fort et al., 2021), and RandAugment (Cubuk et al., 2020) image augmentations (Wightman, 2019). An ablation study on the hyperparameters introduced by CosPress is provided in Section C of the supporting information.

Following the DINOv2 kNN evaluation (Wu et al., 2018) and linear probing approach (Oquab et al., 2024) with an additional batchnorm layer (Lee et al., 2023), evaluations are undertaken on the ImageNet validation set, as well as nine fine-grained classification benchmarks (Oxford Pets (Parkhi et al., 2012), FGVC Aircraft (Maji et al., 2013), Describable Textures (Cimpoi et al., 2014), Stanford Cars (Krause et al., 2013), CUB200 (Wah et al., 2011), CIFAR-10/100 (Krizhevsky et al., 2009), Flowers-102 (Nilsback & Zisserman, 2008) and Food-101 (Bossard et al., 2014)) and the Pascal VOC 2012 segmentation task (Everingham et al., 2012). Performance is also tested on several robustness and generalisation benchmarks including ImageNet-V2

(Recht et al., 2019), Sketch (Wang et al., 2019), ImageNet-R (Hendrycks et al., 2021a) and ImageNet-A (Hendrycks et al., 2021b).

We additionally consider the OpenOOD benchmarks (Yang et al., 2022a). Foundation models are trained on a diverse dataset and excel in this task, and whether distilled students can reproduce this performance has not been previously considered. This section focuses on the ImageNet-1K OpenOOD benchmark, which uses SSB-hard (Bitterwolf et al., 2023) and NINCO (Vaze et al., 2022) as near OOD data, and iNaturalist (Van Horn et al., 2018), OpenImage-O (Wang et al., 2022) and Describable Textures (Cimpoi et al., 2014) as far OOD data.

### 4.2 Results

Table 2: **ImageNet classification.** Comparison of performance on ImageNet-1K under kNN and linear probing evaluation approaches. We report the mean and standard deviation over four runs with different random seeds for the Proteus and CosPress ViT-Ti/14 models.

| Method | Arch | Teacher | kNN | Linear |
|---|---|---|---|---|
| Proteus | ViT-Ti/14 | DINOv2 ViT-S/14 | $73.0 \pm 0.1$ | $76.1 \pm 0.2$ |
| Proteus-$V_kD$ | ViT-Ti/14 | DINOv2 ViT-S/14 | 73.0 | 75.9 |
| CosPress | ViT-Ti/14 | DINOv2 ViT-S/14 | $74.3 \pm 0.1$ | $76.6 \pm 0.1$ |
| | | DINOv2 ViT-S/14 | 79.0 | 81.1 |
| Proteus | ViT-S/14 | DINOv2 ViT-B/14 | 79.8 | 82.0 |
| CosPress | ViT-S/14 | DINOv2 ViT-B/14 | **80.4** | **82.3** |

Table 3: **Distillation components.** Results for kNN evaluations on different components of the distillation process for models distilled on ImageNet-1K. For Proteus, results are shown for the class token student head.

| Method | Arch | Teacher | kNN | | | |
|---|---|---|---|---|---|---|
| | | *DINOv2* | Backbone | Stu. head | Tea. head | Teacher |
| Proteus | ViT-Ti/14 | ViT-S/14 | 73.1 | 73.5 | | 79.0 |
| Proteus-$V_kD$ | ViT-Ti/14 | ViT-S/14 | 73.0 | 73.3 | | 79.0 |
| CosPress | ViT-Ti/14 | ViT-S/14 | **74.3** | | 78.8 | 79.0 |
| Proteus | ViT-S/14 | ViT-B/14 | 79.8 | 80.0 | | 82.1 |
| CosPress | ViT-S/14 | ViT-B/14 | **80.4** | | 82.1 | 82.1 |

**CosPress trains more competitive students.** Table 2 shows that CosPress trains students with better performance in comparison to Proteus (Zhang et al., 2025), for both the linear probing and kNN evaluation methods. These improvements are statistically significant, taking into account the low variability observed across different random seeds. It is also found that the teacher head can project the embeddings from the teacher network into the latent space of the student with minimal loss of kNN accuracy, and that the token student head from Proteus has a higher kNN accuracy than the model backbone (Table 3). These observations confirm the motivations for CosPress—the Proteus student heads are not an uninformative mapping into a higher dimensional space, but are contaminated with information relevant for reproducing the teacher model. Further, a high-quality projection that compresses the teacher emebeddings into the latent space of the student—that preserves cosine similarity—can be learnt, and this provides more effective supervision.

In Table 2 we also consider Proteus-$V_kD$, where the projection matrices $\mathbf{W}$ in the Proteus student head are constrained to be right-orthogonal using the $V_kD$ approach (Miles et al., 2024). This method builds a re-parametrisation map using skew symmetry and a matrix exponential approximation to construct $\mathbf{W}$ such that it is approximately right-orthogonal. Table 2 shows that in this context, this re-parametrisation does not significantly impact performance, and does not completely prevent contamination of the student head $g_\phi$.

It is also observed in Table 2 that the CosPress and Proteus approaches can outperform the distilled DINOv2 models of the same size. However, it is challenging to determine if these distillation approaches are more

effective, as the DINOv2 models were trained on a larger proprietary dataset, of which the ImageNet-1K dataset was only a small subset.

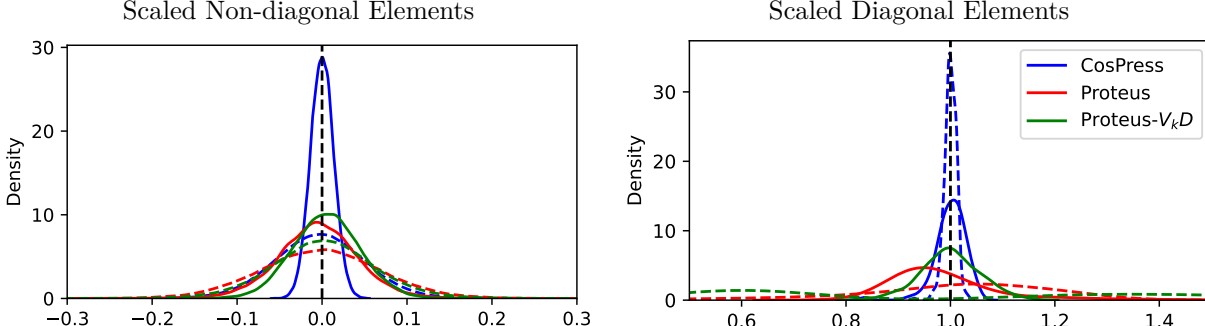

Figure 3: **Visualising orthogonality.** Kernel density estimate plots of the diagonal and non-diagonal elements for the scaled Gram matrices of the linear maps $\mathbf{W}$ in the teacher and student heads, drawn from CosPress and Proteus respectively. The dashed coloured lines represent $\mathbf{W}^\top \mathbf{W}/\alpha$ and the solid lines represent $\mathbf{W}\mathbf{W}^\top/\beta$, where $\alpha, \beta$ are defined as in Table 4. A perfectly orthogonal matrix $\mathbf{W}$ will have a Gram matrix with density on the black dashed vertical lines.

Table 4: **Measuring orthogonality.** Distance measures of the scaled Gram matrices $\mathbf{A} = \mathbf{W}^\top \mathbf{W}/\alpha$ and $\mathbf{B} = \mathbf{W}\mathbf{W}^\top/\beta$ for the projection matrix in the Proteus student and CosPress teacher head, and their respective identity matrices $\mathbf{I}_{D_T}, \mathbf{I}_{D_S}$ of the same dimensions. For each matrix $\alpha, \beta$ is set to the mean of the diagonal elements of $\mathbf{A}, \mathbf{B}$, which minimises error under the Frobenius norm.

| Method | Distance to the Identity Measures | | | |
|---|---|---|---|---|
| | $\|\mathbf{A} - \mathbf{I}_{D_T}\|_F$ | $\|\mathbf{B} - \mathbf{I}_{D_S}\|_F$ | $\mathrm{Tr}\left(\|\mathbf{A} - \mathbf{I}_{D_T}\|\right)$ | $\mathrm{Tr}\left(\|\mathbf{B} - \mathbf{I}_{D_S}\|\right)$ |
| Proteus | 27.3 | 9.5 | 57.1 | 17.0 |
| Proteus-$V_kD$ | 24.9 | 7.7 | 152.7 | 8.2 |
| CosPress | **20.3** | **2.6** | **3.4** | **4.2** |

**CosPress learns an approximately left and right orthogonal projection.** As theorised in Lemma 1, it is found that CosPress learns a linear map $\mathbf{W}$ in the teacher head (Eq. (13)) that is approximately left and right-orthogonal, up to a scaling factor. More concisely, we find that

$$\mathbf{W}^\top \mathbf{W}/\alpha \approx \mathbf{I}, \qquad \mathbf{W}\mathbf{W}^\top/\beta \approx \mathbf{I} \tag{16}$$

where $\alpha, \beta$ are positive real numbers. Qualitatively, this can be seen in Fig. 3 where kernel density plots are shown of the elements from Gram matrices formed using the linear maps $\mathbf{W}, \mathbf{W}^\top$. These projections are taken from the CosPress teacher head and the Proteus class token student head obtained while training the ViT-Ti/14 student network. The scaled CosPress Gram matrices are much closer to the identity matrix, and this is measured quantitatively using the Frobenius norm and trace in Table 4.

While the Proteus-$V_kD$ approach in Fig. 3 and Table 4 does learn a projection $\mathbf{W}$ that is approximately right-orthogonal, there is a large degree of error. This is due to the approximation of the matrix exponential that is used in the $V_kD$ method (Miles et al., 2024), and CosPress is able to learn a right-orthogonal projection matrix with less error.

**CosPress improves performance on classification tasks.** Table 5 shows that the models distilled by CosPress have improved or similar performance over Proteus for all downstream fine-grained classification tasks. Competitive accuracy is also achieved with the distilled DINOv2 models of the same size, which CosPress outperforms on six of the nine datasets. Poorest performance is achieved on the FGVC-Aircraft dataset, which likely reflects differences in the training data used for distillation. The LVD-142M dataset used to train and distill the DINOv2 models contains a million images with high similarity to the FGVC-Aircraft

Table 5: **Fine-grained classification.** Comparison of performance on fine-grained classification tasks using a linear probe evaluation.

| Method | Arch | Teacher | Dataset | | | | | | | | | |
|--------|------|---------|-----|------|------|-----|-----|------|------|------|---------|---------|
| | | | C10 | C100 | Food | CUB | DTD | Pets | Cars | Aircr | Flowers | **Average** |
| Proteus | ViT-Ti/14 | DINOv2 ViT-S/14 | **95.1** | 81.4 | 83.5 | 84.1 | 72.9 | **94.2** | 72.8 | 54.1 | 96.0 | 81.6 |
| CosPress | ViT-Ti/14 | DINOv2 ViT-S/14 | 94.9 | **81.9** | **84.6** | **85.1** | **73.8** | 94.1 | **75.3** | **55.7** | **96.8** | **82.5** |
| | | DINOv2 ViT-S/14 | 97.7 | 87.5 | 89.1 | 88.1 | 80.6 | 95.1 | 81.6 | 74.0 | 99.6 | 88.1 |
| Proteus | ViT-S/14 | DINOv2 ViT-B/14 | **97.8** | **87.7** | 89.7 | 88.4 | **78.0** | **95.9** | 82.8 | 62.9 | 97.6 | 86.8 |
| CosPress | ViT-S/14 | DINOv2 ViT-B/14 | **97.8** | 87.6 | **90.3** | **88.9** | **78.0** | **95.9** | **84.0** | **63.4** | **98.8** | **87.2** |

dataset (Oquab et al., 2024), whereas ImageNet only contains a single *airliner* class with approximately 1300 images.

Table 6: **Semantic segmentation.** Comparison of performance on the Pascal VOC 2012 semantic segmentation task using a linear probe.

| Method | Arch | Teacher | mIoU |
|--------|------|---------|------|
| Proteus | ViT-Ti/14 | DINOv2 ViT-S/14 | 70.5 |
| Proteus w/o patch loss | ViT-Ti/14 | DINOv2 ViT-S/14 | 69.7 |
| CosPress | ViT-Ti/14 | DINOv2 ViT-S/14 | **71.1** |
| | | DINOv2 ViT-S/14 | 81.2 |
| Proteus | ViT-S/14 | DINOv2 ViT-B/14 | 77.3 |
| Proteus w/o patch loss | ViT-S/14 | DINOv2 ViT-B/14 | 77.1 |
| CosPress | ViT-S/14 | DINOv2 ViT-B/14 | **77.9** |

**CosPress improves segmentation performance.** Table 6 shows that CosPress also improves accuracy on downstream segmentation tasks in comparison to Proteus. CosPress does not include the masked patch loss objective, that we confirm improves the performance of Proteus on dense tasks, and incorporating it into CosPress may improve performance further. The DINOv2 distilled model outperforms CosPress in this case. Further pretraining with an increased image resolution was found to be key to improving the performance of the DINOv2 models on dense tasks (Oquab et al., 2024), but is not undertaken in training the CosPress and Proteus student models.

Table 7: **Robustness and generalisation.** Comparison of performance on ImageNet-1K robustness and generalisation benchmarks.

| Method | Arch | Teacher | Test Dataset | | | |
|--------|------|---------|-------|--------|------|------|
| | | *DINOv2* | IN-V2 | Sketch | IN-R | IN-A |
| Proteus | ViT-Ti/14 | ViT-S/14 | 64.3 | 25.5 | 37.8 | 11.4 |
| CosPress | ViT-Ti/14 | ViT-S/14 | **64.9** | **27.9** | **40.7** | **13.2** |
| | | ViT-S/14 (Oquab et al., 2024) | 70.9 | 41.2 | 53.7 | 33.5 |
| Proteus | ViT-S/14 | ViT-B/14 | 72.2 | 38.4 | 50.0 | 29.6 |
| CosPress | ViT-S/14 | ViT-B/14 | **72.5** | **40.4** | **52.3** | **31.5** |

**CosPress distills a more robust student model.** Table 7 shows that CosPress results in improved performance over Proteus across a range of ImageNet-1K robustness and generalisation benchmarks. The DINOv2 distilled model obtains better performance in this instance for all benchmarks except ImageNet-V2, but CosPress closes the gap between the ImageNet-1K and LVD-142M distilled models significantly.

**CosPress reproduces the OOD detection performance of the teacher.** CosPress is faithful to the teacher networks when it comes to OOD detection performance, as shown in Table 1. Proteus performs very poorly on this benchmark, with worse performance observed for larger student models. In contrast, CosPress is able to distill models that have strong OOD performance, even outperforming their DINOv2 counterparts.

Table 8: **Feature distillation with different teachers.** Comparison of performance on ImageNet-1K under kNN and linear probing evaluation methods with other kinds of teacher backbones, using different architectures and training approaches.

| Method | Arch | Teacher | kNN | Linear |
|--------|------|---------|-----|--------|
| Proteus | ViT-T/14 | DINOv2 ViT-B/14 w/reg | 71.1 | 75.1 |
| CosPress | ViT-T/14 | DINOv2 ViT-B/14 w/reg | **74.0** | **76.4** |
| Proteus | ViT-T/16 | CLIP ViT-B/16 | 63.6 | 71.4 |
| CosPress | ViT-T/16 | CLIP ViT-B/16 | **64.0** | **72.0** |

**CosPress improves performance across other teacher networks.** Table 7 demonstrates that Cos-Press also trains higher performing student networks than Proteus when using CLIP (Radford et al., 2021) and DINOv2 w/reg (Darcet et al., 2024) teacher networks. These experiments employ the same hyperparameters as those in Table 2. Additional results exploring feature distillation with ViT-T students and larger teacher networks are provided in Section B of the supporting information.

Table 9: **Training time.** Comparison of training time on ImageNet for 300 epochs with a batch size of 1024 using Nvidia A100 GPUs.

| Method | Arch | Teacher | GPUs | GPU hours | GPU memory |
|--------|------|---------|------|-----------|------------|
| Proteus | ViT-Ti/14 | DINOv2 ViT-S/14 | 1 | 92 | 55GB |
| CosPress | ViT-Ti/14 | DINOv2 ViT-S/14 | 1 | 95 | 47GB |
| Proteus | ViT-S/14 | DINOv2 ViT-B/14 | 2 | 182 | 111GB |
| CosPress | ViT-S/14 | DINOv2 ViT-B/14 | 2 | 154 | 81GB |

**CosPress does not require additional computational resources.** Table 9 provides timings and GPU memory usage for fitting the Proteus and CosPress models described in this section. Training time is similar for the ViT-Ti/14 and DINOv2 ViT-S/14 student-teacher pair, but CosPress is more efficient for larger models. This is due to the masked patch loss in Proteus, which requires that the student network is evaluated once on unmasked inputs, and a second time on masked inputs. As a result, training is faster for CosPress with larger students. CosPress is also slightly more memory efficient compared to Proteus, and further computational savings could be made by freezing the teacher head $h_\phi$ once a sufficiently high quality map has been learned.

# 5 Experiments: Specialist models

This section explores the potential for CosPress feature distillation to improve the performance of specialised models that solve one particular task (e.g. classifying images of food). We refer to this process, where an additional feature distillation training step is undertaken on a target dataset, as CosPress finetuning. This approach can train highly performant small networks, that also have improved results on generalisability and OOD detection benchmarks.

## 5.1 Experimental setup

The CosPress models distilled in the previous section are compared with models that have been further finetuned with CosPress—an additional pretraining step where distillation is undertaken on a smaller target dataset of interest. The strong DeiT (Touvron et al., 2021) pretrained weights are also considered, which were distilled from ImageNet-1K with a larger CNN network using class-based knowledge distillation (Hinton et al., 2015). The same hyperparameters and training methodology is used as in Section 4, with the exception of the number of training and warmup epochs.

This section focuses on a set of small-scale tasks, including CIFAR-10/100 (Krizhevsky et al., 2009), Food-101 (Bossard et al., 2014) and Oxford Pets (Parkhi et al., 2012). We employ 300 training epochs and 10 warmup epochs for CIFAR-10/100, and 3000 training epochs and 100 warmup epochs for Oxford Pets. The

DINOv2 linear probe evaluation method is employed (Oquab et al., 2024), as well as finetuning using the DeiT recipe (Touvron et al., 2021). When training models with this latter approach, the linear prediction head is trained before finetuning the backbone, to avoid distorting the pretrained features (Kumar et al., 2022).

## 5.2 Results

Table 10: **Specialist models — accuracy.** Comparison of performance on fine-grained image classification tasks.

| Method | Arch | Teacher | Pretraining dataset | Linear | | | | DeiT | | | |
|---|---|---|---|---|---|---|---|---|---|---|---|
| | | | | C10 | C100 | Food | Pets | C10 | C100 | Food | Pets |
| DeiT (Touvron et al., 2021) | ViT-Ti/16 | RegNetY-16GF | ImageNet | 93.1 | 77.7 | 77.7 | 93.3 | 98.3 | 87.8 | 89.9 | **93.0** |
| CosPress | ViT-Ti/14 | DINOv2 ViT-S/14 | ImageNet | 94.9 | 81.9 | 84.6 | 94.1 | 98.7 | 89.0 | 91.7 | 92.7 |
| CosPress | ViT-Ti/14 | DINOv2 ViT-S/14 | ImageNet → Target dataset | **97.6** | **86.3** | **89.8** | **94.9** | **98.8** | **89.6** | **92.7** | **93.0** |

**CosPress finetuning improves the performance of specialist models.** Table 10 shows that CosPress finetuning improves downstream performance, even when the training datasets are quite small. For every dataset considered, this additional pretraining step improves linear probe evaluations with a frozen backbone by a significant margin (1-5%). These benefits remain under the strong DeiT training recipe, which further finetunes the model backbone. While CIFAR-10/100 and Food-101 have much stronger results under DeiT finetuning, we find that Oxford Pets has best performance with a linear probe evaluation after CosPress finetuning. This reflects the small size of the Oxford pets dataset, which makes training ViT networks challenging.

Table 11: **State-of-the-art lightweight models.** Comparison of best CosPress models to other approaches for training state-of-the-art lightweight models for specialised tasks.

| Method | Architecture | Parameters | Dataset | | | |
|---|---|---|---|---|---|---|
| | | | C10 | C100 | Food | Pets |
| NAT (Lu et al., 2021) | MobileNetV2 (Sandler et al., 2018) | 4.5-9.0M | 98.4 | 88.3 | 89.4 | 94.3 |
| CeiT (Yuan et al., 2021) | CeiT-T | 6.4M | 98.5 | 88.4 | | 93.8 |
| CosPress | ViT-Ti/14 | 5.5M | **98.8** | **89.6** | **92.7** | **94.9** |

A ViT-Tiny network finetuned with CosPress can have competitive accuracy compared to other approaches in the literature that have been highly optimised to perform well on specialist tasks with a small and efficient model. Table 11 shows that CosPress finetuning trains competitive networks in comparison to Neural Architecture Transfer (NAT) (Lu et al., 2021) and Convolution-enhanced image Transformers (CeiT) (Yuan et al., 2021). Feature distillation methods like CosPress are an additional approach, that could be used in conjunction with these techniques to build highly performant lightweight vision models.

Table 12: **Specialist models — generalisability.** Comparison of generalisability of specialist models on the cartoon subsets of the CIFAR-10-W benchmark (Sun et al., 2024). We report mean per-class accuracy due to dataset imbalances. In-distribution training images (top) and cartoon images (bottom) are included for reference.

| Method | Arch | Teacher | Pretraining dataset | Linear | | | | DeiT | | | |
|---|---|---|---|---|---|---|---|---|---|---|---|
| | | | | Diff | Bin | Bai | 360 | Diff | Bin | Bai | 360 |
| DeiT (Touvron et al., 2021) | ViT-Ti/16 | RegNetY-16GF | ImageNet | 65.9 | 51.0 | 47.6 | 48.8 | 86.9 | 62.6 | 56.0 | 60.1 |
| CosPress | ViT-Ti/14 | DINOv2 ViT-S/14 | ImageNet | 70.8 | 49.5 | **48.7** | **50.3** | 88.5 | 63.2 | 56.3 | 60.7 |
| CosPress | ViT-Ti/14 | DINOv2 ViT-S/14 | ImageNet → Target dataset | **73.4** | **52.5** | 48.6 | 49.9 | **89.1** | **64.1** | **57.5** | **61.4** |

**CosPress finetuning improves the generalisability of specialist models.** The challenging cartoon subsets of the CIFAR-10-W benchmark (Sun et al., 2024) are used to test generalisation performance on CIFAR-10. Table 12 shows that CosPress finetuning leads to improved generalisability for specialist models on CIFAR-10. Under a linear probing evaluation, CosPress finetuning strongly improves generalisability on two of the four datasets, and improves generalisability for all datasets even after DeiT finetuning.

Table 13: **Specialist models — OOD detection.** Comparison of performance on the OpenOOD benchmark (Yang et al., 2022a). The AUC is reported for detecting OOD images.

| Method | Arch | Teacher | Pretraining dataset | Frozen backbone | | | | DeiT finetuned | | | |
|---|---|---|---|---|---|---|---|---|---|---|---|
| | | | | CIFAR-10 | | CIFAR-100 | | CIFAR-10 | | CIFAR-100 | |
| | | | | Near-OOD | Far-OOD | Near-OOD | Far-OOD | Near-OOD | Far-OOD | Near-OOD | Far-OOD |
| DeiT (Touvron et al., 2021) | ViT-Ti/16 | RegNetY-16GF | ImageNet | 57.01 | 47.04 | 58.14 | 46.19 | 96.79 | 98.69 | 87.45 | 86.73 |
| CosPress | ViT-Ti/14 | DINOv2 ViT-S/14 | ImageNet | 93.44 | 95.87 | 85.23 | 76.37 | 96.69 | 98.59 | 87.90 | **89.87** |
| CosPress | ViT-Ti/14 | DINOv2 ViT-S/14 | ImageNet → Target dataset | **95.12** | **98.02** | **87.00** | **80.95** | **97.05** | **98.79** | **89.52** | 89.53 |

**CosPress finetuning improves the OOD detection performance of specialist models.** The CIFAR-10/100 OpenOOD benchmarks (Yang et al., 2022a) are used to test OOD detection performance. Table 13 shows that CosPress finetuning leads to improved performance on the OpenOOD (Yang et al., 2022a) benchmark for specalist models on CIFAR-10/100. Without DeiT finetuning, strong improvements in OOD detection are observed under CosPress finetuning over the ImageNet pretrained baselines. Some improvements remain after DeiT finetuning, but are smaller.

## 6 Discussion

**Significance of angles in deep learning.** Language models have been shown to exhibit a principle of superposition, wherein concepts are encoded along nearly orthogonal directions in representation space (Bricken et al., 2023). While only $d$ vectors can be exactly orthogonal in a $d$-dimensional space, high-dimensional geometry allows for the construction of up to $\exp(d)$ approximately orthogonal vectors (with pairwise cosine similarity less than $\epsilon > 0$) enabling the representation of a vastly larger set of concepts in practice (Elhage et al., 2022). This phenomenon is closely related to the Johnson-Lindenstrauss lemma (Freksen, 2021), and similar properties have been observed in foundation models for computer vision (Bhalla et al., 2024). By preserving angular relationships between image embeddings, CosPress maintains the semantic structure of the foundation model feature space in the student networks it trains.

**Limitations.** Even with CosPress, a small generalisation gap remains. Models trained with CosPress do not generalise quite as well as the original DINOv2 distilled variants (Table 7). Without access to the proprietary LVD-142M dataset, it is difficult to determine whether this gap arises from the limitation of performing feature distillation solely on ImageNet-1K, or from a shortcoming in the methodology itself.

## 7 Conclusion

This paper introduces CosPress, a feature distillation approach designed to train highly performant student networks from a foundation model teacher with a Vision Transformer (Dosovitskiy et al., 2021) architecture, that reproduces their properties in regards to generalisation, robustness and OOD detection. This is achieved by introducing a teacher head, that maps from the higher dimensional latent space of the teacher network into the smaller dimensional space of the student, and training this mapping to preserve the cosine similarity of images within these embedding spaces. CosPress trains a faithful student, that more closely replicates the behaviour of the teacher network in comparison to the Proteus approach (Zhang et al., 2025), where a student head is used to align the student outputs with the teacher.

## Acknowledgements

We acknowledge the Traditional Custodians of the unceded land on which the research detailed in this paper was undertaken, the Wurundjeri Woi Wurrung and Bunurong peoples of the Kulin nation, and pay

our respects to their Elders past and present. This research was undertaken using the LIEF HPC-GPGPU Facility hosted at the University of Melbourne. This Facility was established with the assistance of LIEF Grant LE170100200. This research was also undertaken with the assistance of resources and services from the National Computational Infrastructure (NCI), which is supported by the Australian Government. Evelyn J. Mannix was supported by an Australian Government Research Training Program Scholarship to complete this work.

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

## Supplementary Material



## A    The Johnson–Lindenstrauss Lemma

The Johnson–Lindenstrauss (JL) Lemma (Freksen, 2021) states that for a set of points $\mathbf{X}$ in a high dimensional space, there exists a function that can map these points into a lower dimensional space, within error $\epsilon$, where this error depends on the dimension of the target space $m$ and the size of the set of points $|\mathbf{X}|$. In its standard form, the JL lemma states that Euclidean distances are preserved.

**Lemma 2** (Johnson–Lindenstrauss; (Freksen, 2021)). *For every $d \in \mathbb{N}_1$, $\epsilon \in (0,1)$ and $\mathbf{X} \subset \mathbb{R}^d$, there exists a function $f : \mathbb{R}^d \to \mathbb{R}^m$ where $m = \Theta(\epsilon^{-2} \log |\mathbf{X}|)$ such that for every $x, y \in \mathbf{X}$,*

$$\left| \|f(x) - f(y)\|_2^2 - \|x - y\|_2^2 \right| \leq \epsilon \|x - y\|_2^2 \tag{17}$$

Morever, the map $f$ can be constructed using a simple approach. Given a matrix $\mathbf{M} \in \mathbb{R}^{m \times d}$ with every element drawn from a standard normal distribution $N(0,1)$, then

$$f(x) := \frac{1}{\sqrt{m}} \mathbf{M} x \tag{18}$$

is a linear map that satisfies Lemma 2 with a probability given by the norm preservation lemma. This function is also referred to as a JL transform.

**Lemma 3** (Norm preservation; (Freksen, 2021)). *Let $\epsilon \in (0,1)$. If $f$ is constructed as above with $m = \Theta(\epsilon^{-2} \log \delta^{-1})$, and $x \in \mathbb{R}^d$ is a unit vector, then*

$$\mathbb{P}\left[ \|f(x)\|_2^2 \in (1 \pm \epsilon) \right] \geq 1 - \delta \tag{19}$$

Again, for target spaces with larger dimension $m$ this Lemma 3 states that it is more likely that a high quality map will be sampled. A similar result also holds for angles (Magen, 2007), which gives

**Lemma 4** (Angles; (Magen, 2007)). *Let $\epsilon < \frac{1}{3}$ and let $n, t$ be integers for which $t > 60\epsilon^{-2} \log n$. Then for any $n$-point subset $\mathbf{X}$ of the Euclidean space $\mathbb{R}^N$, there is a linear contracting embedding $f(\mathbf{X}) \to \mathbb{R}^t$, under which angles are preserved to within a (double-sided) factor of $1 + 8/\pi\sqrt{\epsilon}$.*

The proof of Lemma 4 also relies on $f$ being generated as a random projection as above (Magen, 2007). The JL transform matrix $\mathbf{M}$ has a further interesting property, as observed in this work, that we refer to as approximate left and right orthogonality.

**Lemma 5** (Approximate left and right orthogonality for JL transforms). *Given a matrix $\mathbf{M} \in \mathbb{R}^{m \times d}$ with elements drawn from $N(0,1)$, there is*

$$\frac{\mathbb{E}\|\mathbf{M}\mathbf{M}^\top - \alpha \mathbf{I}_m\|_F^2}{\mathbb{E}\|\mathbf{M}\mathbf{M}^\top\|_F^2} = \frac{m}{d+m}, \qquad \frac{\mathbb{E}\|\mathbf{M}^\top \mathbf{M} - \beta \mathbf{I}_m\|_F^2}{\mathbb{E}\|\mathbf{M}^\top \mathbf{M}\|_F^2} = \frac{d}{d+m}.$$

*Proof.* Using the properties of the Wishart distribution, which given the construction of $\mathbf{M}$, we have

$$\mathbf{M}\mathbf{M}^\top \sim W_m(\mathbf{I}_m, d) \tag{20}$$

$$\mathbf{M}^\top \mathbf{M} \sim W_d(\mathbf{I}_d, m), \tag{21}$$

from which we have the following results for the mean and variance

$$E[\mathbf{M}\mathbf{M}^\top] = d\mathbf{I}_m, \operatorname{Var}([\mathbf{M}\mathbf{M}^T]_{ij}) = d \tag{22}$$

$$E[\mathbf{M}^\top \mathbf{M}] = m\mathbf{I}_d, \operatorname{Var}([\mathbf{M}^T \mathbf{M}]_{ij}) = m. \tag{23}$$

These expectations imply Lemma 5. $\qquad \square$

Morever, this property of approximate left and right orthogonality does not just apply to JL transformations, but any linear map $\mathbf{M} \in \mathbb{R}^{m \times d}$ with $d > m$ such that $\mathbf{M}$ is approximately left-orthogonal with $\mathbf{M}^\top \mathbf{M} \approx \mathbf{I}_d$. $\mathbf{M}$ cannot be exactly left-orthogonal, as the rank of $\mathbf{M}$ and $\mathbf{M}^\top$ are both at most $m$, while $\mathbf{I}_d$ is of rank $d$ which is greater than $m$.

**Lemma 1.** *(Approximate left-orthogonality implies right-orthogonality) For any matrix $\mathbf{M} \in \mathbb{R}^{m \times d}$ with $m < d$ with rank $m$, we have the inequality $\|\mathbf{M}\mathbf{M}^\top - \frac{d}{m}\mathbf{I}_m\|_F \le \|\mathbf{M}^\top \mathbf{M} - \mathbf{I}_d\|_F$. The converse inequality does not hold.*

*Proof.* We can write $\mathbf{M}^\top \mathbf{M} = \mathbf{I}_d + \mathbf{E}$, which provides that

$$\|\mathbf{M}^\top \mathbf{M} - \mathbf{I}_d\|_F = \|\mathbf{E}\|_F \tag{24}$$

where $\mathbf{E}$ is an error matrix. Let us sample $m$ columns of $\mathbf{M}$ to produce a square matrix $\mathbf{K} \in \mathbb{R}^{m \times m}$. This provides for $\mathbf{K}$, that we have $\mathbf{K}^\top \mathbf{K} = I_m + \mathbf{E}_K$ where $\mathbf{E}_K$ are the same columns sampled from the error matrix $\mathbf{E}$. This means that $\|\mathbf{E}_K\|_F \le \|\mathbf{E}\|_F$.

We then introduce the singular value decomposition of $\mathbf{K} = \mathbf{U}\boldsymbol{\Sigma}\mathbf{V}^\top$, where $\mathbf{U}$ and $\mathbf{V}$ are orthonormal matrices and $\boldsymbol{\Sigma}$ is a square diagonal matrix containing the singular values of $\mathbf{K}$, to give

$$\mathbf{K}^\top \mathbf{K} = \mathbf{V}\boldsymbol{\Sigma}^\top \boldsymbol{\Sigma} V^\top = \mathbf{I}_m + \mathbf{E}_K \tag{25}$$

$$\boldsymbol{\Sigma}^\top \boldsymbol{\Sigma} = \mathbf{V}^\top (\mathbf{I}_m + \mathbf{E}_K) V \tag{26}$$

$$\boldsymbol{\Sigma}^\top \boldsymbol{\Sigma} = \mathbf{I}_m + \mathbf{V}^\top \mathbf{E}_K \mathbf{V} \tag{27}$$

which we can use, as $\boldsymbol{\Sigma}^\top \boldsymbol{\Sigma} = \boldsymbol{\Sigma}\boldsymbol{\Sigma}^\top$ for a square diagonal matrix, to find that

$$\mathbf{K}\mathbf{K}^\top = \mathbf{U}\boldsymbol{\Sigma}\boldsymbol{\Sigma}^\top \mathbf{U}^\top \tag{28}$$

$$= \mathbf{U}(\mathbf{I}_m + \mathbf{V}^\top \mathbf{E}_K \mathbf{V})\mathbf{U}^\top \tag{29}$$

$$= \mathbf{I}_m + \mathbf{U}\mathbf{V}^\top \mathbf{E}_K \mathbf{V}\mathbf{U}^\top \tag{30}$$

Let us consider a sample of $d$ sets of $m$ columns, such that each of the $d$ columns is selected $m$ times to produce a set of $\mathbf{K}_l$ matrices where $l \in \{1, ..., d\}$ without repeated columns. Then we observe that,

$$\left[\mathbf{M}\mathbf{M}^\top\right]_{i,j} = \sum_{k=1}^d \mathbf{M}_{ik}\mathbf{M}_{kj} \tag{31}$$

$$= \frac{1}{m}\sum_{l=1}^m \sum_{k=1}^d \mathbf{M}_{ik}\mathbf{M}_{kj} \tag{32}$$

$$= \frac{1}{m}\sum_{l=1}^d \sum_{k=1}^m [\mathbf{K}_l]_{ik}[\mathbf{K}_l]_{kj} \tag{33}$$

as we have sampled our set of $\mathbf{K}_l$ matrices from the columns of $\mathbf{M}$ such that each $\mathbf{M}_{ik}\mathbf{M}_{kj}$ element in the sum of their products occurs $m$ times. This gives

$$\mathbf{M}\mathbf{M}^\top = \frac{1}{m}\sum_{l=1}^d \mathbf{K}_l \mathbf{K}_l^\top \tag{34}$$

and introducing Eq. (30) from above obtains

$$\mathbf{M}\mathbf{M}^\top = \frac{1}{m}\sum_{l=1}^d \left(\mathbf{I}_m + \mathbf{U}\mathbf{V}^\top \mathbf{E}_{K_l}\mathbf{V}\mathbf{U}^\top\right) \tag{35}$$

$$= \frac{d}{m}\mathbf{I}_m + \frac{1}{m}\sum_{l=1}^d \mathbf{U}\mathbf{V}^\top \mathbf{E}_{K_l}\mathbf{V}\mathbf{U}^\top \tag{36}$$

---

**Algorithm 1** Algorithm for feature distillation with CosPress.

---

**Input:** Training set $X$, teacher model $T$, student model $S_\theta$, teacher head $h_\phi$, $N_E$ number of epochs, Aug(.) augmentation strategy

Randomly initialise student model $S_\theta$ and teacher head $h_\phi$, or initialise using previous weights if fine-tuning;

$i = 0$;

**while** $i < N_E$ **do**

    Randomly split $X$ into $B$ mini-batches;

    **for** $x_b \in \{X_1, ..., X_b, ..., X_B\}$ **do**

        Generate augmented views: $\mathbf{X} = \text{Aug}(x_b)$;

        Compute dimensionality reduction objective (Eq. (12)):

        $\mathcal{L}_{\text{dim-red}}(\mathbf{X}; \phi) = L_{\text{KL}}(h_\phi(T^c(\mathbf{X})), T^c(\mathbf{X})) + \frac{1}{|\mathbf{X}|} \sum_i L_{\text{KL}}(h_\phi(T(x_i)), T(x_i))$

        Compute student objective, while freezing $\phi$ (Eq. (15)):

        $\mathcal{L}_{\text{student}}(\mathbf{X}; \theta) = L_{\text{cosine}}(S_\theta^c(\mathbf{X}), h_{\phi_{\text{frozen}}}(T^c(\mathbf{X}))) + L_{\text{cosine}}(S_\theta(\mathbf{X}), h_{\phi_{\text{frozen}}}(T(\mathbf{X})))$

        Combine losses: $\mathcal{L} = \mathcal{L}_{\text{student}}(\mathbf{X}; \theta) + \mathcal{L}_{\text{dim-red}}(\mathbf{X}; \phi)$;

        Minimise loss $\mathcal{L}$ by updating parameters of $\theta$ and $\phi$;

    **end for**

    $i = i + 1$;

**end while**

---

which provides that

$$\|\mathbf{M}\mathbf{M}^\top - \frac{d}{m}\mathbf{I}_m\|_F = \|\frac{1}{m}\sum_{l=1}^d \mathbf{U}\mathbf{V}^\top\mathbf{E}_{K_l}\mathbf{V}\mathbf{U}^\top\|_F \tag{37}$$

$$\leq \frac{d}{m}\|\mathbf{E}\|_F \tag{38}$$

which proves the first statement. It is easy to see the converse inequality is not true, simply by constructing a matrix $\mathbf{M} \in \mathbb{R}^{d \times m}$ with the first $m$ rows equal to the identity. This matrix satisfies that $\mathbf{M}\mathbf{M}^\top = \mathbf{I}_m$, but $\mathbf{M}^\top\mathbf{M}$ will have $d - m$ rows with only zeros in them. This proves the second statement. $\qquad\square$

# B  Further Results

Table S14: **ImageNet classification — larger teachers.** Comparison of performance on ImageNet-1K under kNN and linear probing evaluation approaches.

| Method | Arch | Teacher | kNN | | | | Linear |
|--------|------|---------|----------|--------------|--------------|---------|--------|
| | | | Backbone | Student head | Teacher head | Teacher | |
| Proteus | ViT-Ti/14 | DINOv2 ViT-S/14 | 73.1 | 73.5 | | 79.0 | 76.1 |
| CosPress | ViT-Ti/14 | DINOv2 ViT-S/14 | **74.3** | | 78.8 | 79.0 | **76.8** |
| Proteus | ViT-Ti/14 | DINOv2 ViT-B/14 | 73.4 | 73.8 | | 82.1 | 76.9 |
| CosPress | ViT-Ti/14 | DINOv2 ViT-B/14 | **75.6** | | 81.9 | 82.1 | **77.9** |

**Larger teachers result in better accuracy but have poorer quality dense features.** We observe that training with larger teachers in comparison to the student requires longer training runs. To achieve best results efficiently, the pretrained models from a smaller teacher are used as a starting point. Then, the student and teacher heads are first trained with a frozen pretrained student model (allowing the CosPress teacher heads to minimize the student loss) for 30 epochs. Finally, the models are trained for 300 epochs using the same distillation approach as previously. Table S14 shows that this improves the performance of the student models, with CosPress seeing larger improvements in accuracy on ImageNet-1K in comparison to Proteus.

However, this results in poorer results in dense image tasks, like semantic segmentation. Table S15 shows that the students trained with a larger teacher network have poorer mIoU for a linear probe on the Pascal VOC 2012 dataset. Undertaking longer distillation runs, or continuing training using a larger image resolution potentially might be helpful in these cases.

Table S15: **Semantic segmentation — larger teachers.** Comparison of performance on the Pascal VOC 2012 semantic segmentation task using a linear probe.

| Method | Arch | Teacher | mIoU |
|--------|------|---------|------|
| Proteus | ViT-Ti/14 | DINOv2 ViT-S/14 | 70.5 |
| CosPress | ViT-Ti/14 | DINOv2 ViT-S/14 | **71.1** |
| Proteus | ViT-Ti/14 | DINOv2 ViT-B/14 | 68.1 |
| CosPress | ViT-Ti/14 | DINOv2 ViT-B/14 | **69.7** |

**Further out-of-distribution detection results.** The OOD detection results for all of the OpenOOD datasets (Yang et al., 2022a) are presented in Table S16, Table S17 and Table S18. The tables report the ROC-AUC for detecting OOD images and the False Positive Rate (FPR) using a 95% threshold for including all in-distribution images. To produce these results, the KNN+ (Sun et al., 2022) OOD metric is used to measure the performance of the model backbones. For ImageNet-1K, we sample 1% of the dataset (12,812 images) and set $k = 10$ to measure distance to OOD samples. For CIFAR-10/100, we sample 100% of the dataset (50,000 images) and set $k = 1$.

Table S16: **Out-of-distribution detection.** Comparison of performance on the OpenOOD benchmark for the ImageNet-1K dataset. The ↑ means larger values are better and the ↓ means smaller values are better.

| Method | Arch | Teacher | Near OOD Datasets | | | | | | Far OOD Datasets | | | | | | | |
|--------|------|---------|------|------|------|------|------|------|------|------|------|------|------|------|------|------|
| | | | SSB-hard | | NINCO | | Average | | iNaturalist | | OpenImage-O | | Textures | | Average | |
| | | | AUROC↑ | FPR↓ | AUROC↑ | FPR↓ | AUROC↑ | FPR↓ | AUROC↑ | FPR↓ | AUROC↑ | FPR↓ | AUROC↑ | FPR↓ | AUROC↑ | FPR↓ |
| Proteus | ViT-Ti/14 | DINOv2 ViT-S/14 | 55.59 | 92.24 | 72.76 | 79.22 | 64.17 | 85.73 | 60.08 | 91.47 | 73.13 | 75.34 | **89.46** | **37.1** | 74.22 | 67.97 |
| CosPress | ViT-Ti/14 | DINOv2 ViT-S/14 | **63.39** | **84.98** | **77.58** | **69.59** | **70.49** | **77.29** | **95.81** | **22.57** | **90.72** | **40.62** | 86.57 | 48.45 | **91.03** | **37.21** |
| | | DINOv2 ViT-S/14 | 65.76 | 81.8 | 79.39 | 66.45 | 72.58 | 74.12 | 98.74 | 4.76 | 92.23 | 35.44 | 87.04 | 48.45 | 92.67 | 29.55 |
| Proteus | ViT-S/14 | DINOv2 ViT-B/14 | 53.78 | 97.4 | 68.61 | 91.71 | 61.19 | 94.56 | 39.89 | 99.5 | 61.7 | 91.11 | 84.19 | 69.73 | 61.92 | 86.78 |
| CosPress | ViT-S/14 | DINOv2 ViT-B/14 | **65.75** | **82.98** | **81.24** | **64.71** | **73.5** | **73.84** | **97.31** | **12.27** | **92.77** | **32.95** | **88.72** | **41.71** | **92.93** | **28.98** |

Table S17: **Specialist models — near OOD detection.** Comparison of performance on the OpenOOD benchmark (Yang et al., 2022a). The ↑ means larger values are better and the ↓ means smaller values are better.

| IDD | Method | Arch | Teacher | Pretraining dataset | Near OOD Datasets | | | | | | Average | |
|---|---|---|---|---|---|---|---|---|---|---|---|---|
| | | | | | CIFAR-10 | | CIFAR-100 | | Tiny ImageNet | | | |
| **CIFAR-100** | | | | | AUROC↑ | FPR↓ | AUROC↑ | FPR↓ | AUROC↑ | FPR↓ | AUROC↑ | FPR↓ |
| Frozen | DeiT | ViT-Ti/16 | RegNetY-16GF | ImageNet | 53.53 | 94.96 | | | 62.75 | 85.44 | 58.14 | 90.2 |
| | CosPress | ViT-Ti/14 | DINOv2 ViT-S/14 | ImageNet | 80.39 | 71.45 | | | 90.07 | 34.94 | 85.23 | 53.2 |
| | CosPress | ViT-Ti/14 | DINOv2 ViT-S/14 | ImageNet → Target dataset | 84.08 | 70.28 | | | 89.91 | 44.15 | 87.0 | 57.22 |
| DeiT finetuned | DeiT | ViT-Ti/16 | RegNetY-16GF | ImageNet | 83.91 | 61.04 | | | 90.98 | 44.29 | 87.45 | 52.66 |
| | CosPress | ViT-Ti/14 | DINOv2 ViT-S/14 | ImageNet | 85.45 | 56.76 | | | 90.34 | 47.36 | 87.9 | 52.06 |
| | CosPress | ViT-Ti/14 | DINOv2 ViT-S/14 | ImageNet → Target dataset | 86.7 | 56.45 | | | 92.35 | 39.37 | 89.52 | 47.91 |
| **CIFAR-10** | | | DINOv2 ViT-S/14 | | 87.97 | 56.05 | | | 91.83 | 29.61 | 89.9 | 42.83 |
| Frozen | DeiT | ViT-Ti/16 | RegNetY-16GF | ImageNet | | | 50.88 | 94.08 | 63.15 | 84.75 | 57.01 | 89.42 |
| | CosPress | ViT-Ti/14 | DINOv2 ViT-S/14 | ImageNet | | | 90.22 | 41.5 | 96.67 | 13.63 | 93.44 | 27.57 |
| | CosPress | ViT-Ti/14 | DINOv2 ViT-S/14 | ImageNet → Target dataset | | | 93.76 | 31.79 | 96.49 | 15.86 | 95.12 | 23.82 |
| DeiT finetuned | DeiT | ViT-Ti/16 | RegNetY-16GF | ImageNet | | | 96.5 | 16.34 | 97.08 | 12.52 | 96.79 | 14.43 |
| | CosPress | ViT-Ti/14 | DINOv2 ViT-S/14 | ImageNet | | | 96.15 | 16.57 | 97.24 | 10.62 | 96.69 | 13.6 |
| | CosPress | ViT-Ti/14 | DINOv2 ViT-S/14 | ImageNet → Target dataset | | | 96.76 | 15.94 | 97.33 | 12.02 | 97.05 | 13.98 |
| | | | DINOv2 ViT-S/14 | | | | 94.08 | 29.27 | 97.59 | 10.28 | 95.83 | 19.77 |

Table S18: **Specialist models — far OOD detection.** Comparison of performance on the OpenOOD benchmark (Yang et al., 2022a). The ↑ means larger values are better and the ↓ means smaller values are better.

| IDD | Method | Arch | Teacher | Pretraining dataset | Far OOD Datasets | | | | | | | | Average | |
|---|---|---|---|---|---|---|---|---|---|---|---|---|---|---|
| | | | | | DTD | | MNIST | | SVHN | | Places365 | | | |
| **CIFAR-100** | | | | | AUROC↑ | FPR↓ | AUROC↑ | FPR↓ | AUROC↑ | FPR↓ | AUROC↑ | FPR↓ | AUROC↑ | FPR↓ |
| Frozen | DeiT | ViT-Ti/16 | RegNetY-16GF | ImageNet | 14.97 | 100.0 | 71.86 | 73.77 | 43.35 | 93.18 | 54.57 | 83.06 | 46.19 | 87.5 |
| | CosPress | ViT-Ti/14 | DINOv2 ViT-S/14 | ImageNet | 33.09 | 99.89 | 97.21 | 11.01 | 76.96 | 87.71 | 98.22 | 7.18 | 76.37 | 51.45 |
| | CosPress | ViT-Ti/14 | DINOv2 ViT-S/14 | ImageNet → Target dataset | 44.46 | 98.56 | 95.78 | 18.94 | 86.17 | 67.34 | 97.39 | 12.91 | 80.95 | 49.44 |
| DeiT finetuned | DeiT | ViT-Ti/16 | RegNetY-16GF | ImageNet | 74.07 | 82.86 | 85.18 | 62.99 | 95.81 | 24.57 | 91.87 | 43.29 | 86.73 | 53.43 |
| | CosPress | ViT-Ti/14 | DINOv2 ViT-S/14 | ImageNet | 83.41 | 68.51 | 87.49 | 56.64 | 96.5 | 21.0 | 92.08 | 37.52 | 89.87 | 45.92 |
| | CosPress | ViT-Ti/14 | DINOv2 ViT-S/14 | ImageNet → Target dataset | 79.51 | 67.44 | 90.24 | 48.05 | 96.12 | 22.84 | 92.27 | 37.61 | 89.53 | 43.98 |
| **CIFAR-10** | | | DINOv2 ViT-S/14 | | 42.46 | 99.8 | 96.25 | 15.68 | 77.75 | 88.13 | 97.89 | 8.48 | 78.58 | 53.02 |
| Frozen | DeiT | ViT-Ti/16 | RegNetY-16GF | ImageNet | 17.03 | 100.0 | 71.36 | 73.15 | 42.03 | 92.25 | 57.74 | 80.58 | 47.04 | 86.49 |
| | CosPress | ViT-Ti/14 | DINOv2 ViT-S/14 | ImageNet | 95.24 | 30.79 | 99.17 | 3.54 | 89.13 | 60.24 | 99.97 | 0.18 | 95.87 | 23.69 |
| | CosPress | ViT-Ti/14 | DINOv2 ViT-S/14 | ImageNet → Target dataset | 98.2 | 7.44 | 98.92 | 4.38 | 95.08 | 33.79 | 99.88 | 0.48 | 98.02 | 11.52 |
| DeiT finetuned | DeiT | ViT-Ti/16 | RegNetY-16GF | ImageNet | 97.86 | 9.33 | 97.52 | 9.69 | 99.67 | 0.71 | 99.7 | 0.96 | 98.69 | 5.17 |
| | CosPress | ViT-Ti/14 | DINOv2 ViT-S/14 | ImageNet | 97.95 | 9.31 | 97.41 | 8.99 | 99.63 | 0.32 | 99.36 | 1.51 | 98.59 | 5.03 |
| | CosPress | ViT-Ti/14 | DINOv2 ViT-S/14 | ImageNet → Target dataset | 97.78 | 11.81 | 98.18 | 7.9 | 99.81 | 0.16 | 99.37 | 2.16 | 98.79 | 5.51 |
| | | | DINOv2 ViT-S/14 | | 97.0 | 19.08 | 98.93 | 4.25 | 90.6 | 55.45 | 99.96 | 0.18 | 96.62 | 19.74 |

## C   Ablation Studies

In this section we modify a number of hyperparameters and component choices for CosPress to investigate how these impact performance. In the tables below the bold parameter sets are the default ones used throughout the rest of the paper.

Table S19: **Ablation study: weighting.** Comparison of kNN performance on ImageNet-1K for CosPress models trained with different weightings $\gamma$ for the dimensionality reduction component. The first row uses the frozen gradient approach described in the paper.

| Method | Arch | Teacher | $\gamma$ | kNN |
|---|---|---|---|---|
| CosPress | ViT-Ti/14 | DINOv2 ViT-S/14 | - | 74.3 |
| CosPress | ViT-Ti/14 | DINOv2 ViT-S/14 | 10 | 74.3 |
| CosPress | ViT-Ti/14 | DINOv2 ViT-S/14 | 100 | 74.2 |

**CosPress dimensionality reduction loss weighting.** We consider the performance of weighting the dimensionality reduction loss in Eq. (6), rather than freezing the gradients of $\phi$ in the student loss. This gives an alternative loss function

$$\mathcal{L}_{\text{CosPress}}(\mathbf{X}; \phi, \theta) = \gamma \mathcal{L}_{\text{dim-red}}(\mathbf{X}; \phi) + \mathcal{L}_{\text{student}}(\mathbf{X}; \theta, \phi) \tag{39}$$

where $\gamma$ is a weighting factor prioritises the dimensionality reduction loss when it is set to be greater than one. Table S19 shows that the approach of weighting or freezing gradients leads to similar results.

Table S20: **Ablation study: metric.** Comparison of kNN performance on ImageNet-1K for CosPress models trained with different metrics for the student loss.

| Method | Arch | Teacher | Loss Metric | kNN |
|---|---|---|---|---|
| CosPress | ViT-Ti/14 | DINOv2 ViT-S/14 | **Cosine distance** | 74.3 |
| CosPress | ViT-Ti/14 | DINOv2 ViT-S/14 | MSE | 74.2 |

**CosPress student loss metric.** Table S20 considers the impact on performance of using different metrics for the student loss $\mathcal{L}_{\text{student}}$ for Eq. (15). It is found that using a cosine distance loss leads to slightly better performance in comparison to a mean squared error loss.

Table S21: **Ablation study: temperature.** Comparison of kNN performance on ImageNet-1K for CosPress models trained with different sets of temperatures $\boldsymbol{\tau}$ for the dimensionality reduction loss.

| Method | Arch | Teacher | $\boldsymbol{\tau}$ | kNN |
|---|---|---|---|---|
| CosPress | ViT-Ti/14 | DINOv2 ViT-S/14 | **[0.01, 0.02, 0.03, 0.04, 0.05, 0.06, 0.07, 0.08, 0.09, 0.10]** | 74.3 |
| CosPress | ViT-Ti/14 | DINOv2 ViT-S/14 | [0.01] | 74.1 |
| CosPress | ViT-Ti/14 | DINOv2 ViT-S/14 | [0.10] | 74.5 |
| CosPress | ViT-Ti/14 | DINOv2 ViT-S/14 | [0.01, 0.10] | 74.3 |

**CosPress dimensionality reduction temperature parameters.** Table S20 shows the performance impact of different sets of temperatures $\boldsymbol{\tau}$ in the dimensionality reduction loss for Eq. (11). It is found that these values have a small impact on performance, with the best set being $\boldsymbol{\tau} = [0.10]$, which obtains slightly better performance that the set of parameters chosen for the paper.

