# OpenReview forum: "Preserving Angles Improves Feature Distillation"
_TMLR — Accepted by TMLR_

### Review · Reviewer_KnZq · 2025-09-12

**Summary Of Contributions:**

The authors present a novel feature distillation method, CosPress, which improves the Proteus method. First, the authors notice that a projection from a student feature space to a teacher feature space in Proteus might not preserve token similarities well. They suggest an improvement that changes the direction of the projection from teacher to student and justify this choice by the Johnson-Lindenstrauss lemma. Second, the authors demonstrate that their modification enhances Proteus in various vision tasks (with a focus on ViT-Ti distilled from DINOv2 ViT-S on ImageNet). Additionally, the authors demonstrate that their method indeed provides the expected conceptual benefit in terms of orthogonality of the projection matrices. Finally, the authors show that CosPress is well-suited for a specialized setting that includes an additional distillation step.

**Strengths**
1. The motivation is clear.
2. The proposed method is theoretically sound.
3. The evaluation was performed for a variety of vision tasks.

**Weaknesses**
1. As I understand, the authors use only one training run to compare CosPress with Proteus. Thus, the statistical significance of the findings is unclear.
2. Text lacks clarity in several places.
3. The experiments explore a limited number of architectures for the student and the teacher.
4. Similarly, the experiments primarily focus on distillation using the ImageNet dataset, without ablations.

**Audience:**

Yes

**Audience Explanation:**

The problem is well-motivated, and the method seems to improve the previous baseline.

**Broader Impact Concerns:**

I don't see major broader impact concerns.

**Claims And Evidence:**

Yes

**Claims Explanation:**

I believe that the bulk of the evidence is sufficient since the authors utilize a wide range of vision tasks in their evaluation. However, I still think that the scope could be improved with experiments on different architectures and datasets. Additionally, I believe that some improvements are rather marginal; therefore, repeating training with different seeds is necessary to evaluate statistical significance.

**Requested Changes:**

1. In Section 3, please clarify which losses are calculated between examples in a batch and which are calculated between patches in an image, both for the CosPress and Proteus methods.
2. Similarly, in Section 3, please define what an "approximately orthogonal matrix" means. Why is this notion related to orthogonality? What is different? Why is Lemma 1 important?
3. In Section 4, please provide training details (augmentations, etc.) and **the number of training runs**.
4. In Section 5, please clarify the details for an additional distillation step. Do the hyperparameters of the initial distillation change? Are the hyperparameters of the additional distillation step different?

---

> ### Author Response · Authors · 2025-10-12
> **Response to Review**
>
> Thank you for your review of our paper. Please see our answers to your points below, and let us know if there is anything we can clarify further.
>
> > W 1. The authors use only one training run to compare CosPress with Proteus and the statistical significance of the findings is unclear.
>
> We have further experiments with multiple training seeds (Table 2), showing that variability is limited and the reported improvements are statistically significant.
>
> > W 2. Lack of clarity in the text.
>
> We have edited key sections of the report to improve clarity.
>
> > W 3. The experiments explore a limited number of architectures.
>
> We have added additional experiments to the paper considering CLIP and DINOv2 w/reg teachers, showing that CosPress also improves over Proteus in these cases (Table 8). We focus on ViT architectures due to the significance of this backbone in the development of computer vision foundation models.
>
> > W 4. The experiments primarily focus on distillation using the ImageNet dataset.
>
> In section 5 we consider distillation on datasets other than ImageNet, including CIFAR-10/100, Oxford Pets and Food-101. An ablation study is also provided in Appendix C.
>
> > RC 1. In Section 3, please clarify which losses are calculated between examples in a batch and which are calculated between patches in an image.
>
> We have added clarification on this point on page 6.
>
> > RC 2. In Section 3, please define what an "approximately orthogonal matrix" means.
>
> We have edited this section to make it clearer, adding a definition for approximate orthogonality, relating it to orthogonality, and more clearly highlighting the importance of Lemma 1.
>
> > RC 3. In Section 4, please provide training details (augmentations, etc.) and the number of training runs.
>
> We have added these additional details.
>
> > RC 4. In Section 5, please clarify the details for the additional distillation step.
>
> We use the same hyperparameters for this additional distillation step, other than changing the number of epochs and warmup epochs for each dataset. We have added these additional details.

---

### Review · Reviewer_pBfZ · 2025-09-18

**Summary Of Contributions:**

This paper proposes CosPress, a knowledge distillation method that aims to preserve the angular geometry of teacher embeddings when compressing large vision transformers into smaller students. The main point is that the authors introduce a teacher head that projects teacher features into the student’s latent space while explicitly preserving cosine similarities between embeddings. The student is then trained to align with these compressed teacher features using a cosine distance loss.

Strengths:
- This paper is well-motivated. The idea of enforcing cosine similarity is interesting and sound.
- The authors proposed many useful losses and techniques: the teacher head and similarity distribution loss are simple but effective.
- The authors performed extensive experiments including classification, robustness, segmentation, and OOD detection shows effectiveness over baselines.

Weaknesses:
- The scope of the paper is limited to ViTs and ImageNet; unclear generalization to other architectures or domains.
- Some metrics has limited improvement over baselines, such as in Table 10, Table 7.

**Audience:**

Yes

**Audience Explanation:**

Knowledge distillation is a well-studied area in machine learning. This paper presents a new method for knowledge distillation.

**Claims And Evidence:**

Yes

**Claims Explanation:**

The experimental results are convincing and systematically support the paper’s claims. The authors compare against strong baselines across multiple tasks, such as classification, segmentation, robustness benchmarks, and OOD detection.

**Requested Changes:**

- Broaden evaluation scope, such as non-ViT or non-self-supervised teacher (e.g., CNN or supervised transformer) to demonstrate generalization of the method.

- Provide more training details and analysis of the cost of similarity distribution matching (pairwise computations).

- Discuss the reason why some of the evaluation only has limited improvement.

---

> ### Author Response · Authors · 2025-10-12
> **Response to Review**
>
> Thank you for your review of our paper. Please see our answers to your concerns below, and let us know if there is anything we can clarify further.
>
> > RC 1. Broaden evaluation scope to demonstrate generalization of the method.
>
> The aim of this work is to reproduce the properties of self-supervised teachers with a ViT architecture, which have underpinned the development of computer vision foundation models over the last few years. To broaden the scope of the paper, we have added additional experiments showing that CosPress also provides improved performance when distilling from CLIP and DINOv2 w/reg backbones (Table 8). The paper also considers distillation on datasets other than ImageNet in section 5.
>
> > RC 2. Provide more training details and analysis of the cost of similarity distribution matching.
>
> We have added further training details to Section 4.1. We provide a comparison of the computational costs of CosPress versus Proteus in Table 9.
>
> > RC 3. Discuss the reason why some of the evaluation only has limited improvement.
>
> Our primary focus is on reproducing the properties of foundation models in smaller student models, particularly in terms of out-of-distribution (OOD) detection and robustness. While Proteus fails to match OOD detection performance, CosPress achieves performance comparable to DINOv2 models (Table 1). Moreover, CosPress reduces the robustness gap between DINOv2 models and Proteus students by 60% (Table 7). We also report experiments on variability across different seeds in Table 2, demonstrating that variability is limited and even small improvements can be statistically significant.

---

### Review · Reviewer_g1HL · 2025-09-29

**Summary Of Contributions:**

# Summary

The paper presents Cospress,an adaptation of distilling foundation models to smaller student models that considers the role of angle preservation explicitly. While previous work (the paper compares mainly with Proteus [https://arxiv.org/abs/2407.10366](https://arxiv.org/abs/2407.10366) ) uses a final up-projection to match the teacher model in it’s latent space (following Fitnets [https://arxiv.org/abs/1412.6550](https://arxiv.org/abs/1412.6550) ) approach, Cospress instead projects the teacher down to the students latent space, while trying maintain cosine similarities between different images (via a SNE inspired regularization loss comprised of the $D_{KL}$ between the distributions of von-mises fisher kernel distances between pairs, at different temperatures). The angle perserving dimensionality reduction is trained 1) the pairs of images within a batch 2) the features (patch and class tokens) within a single image. A small analysis is done to justify this approach, drawing links to JL random projection theory and the recent studies of feature superposition is drawn, since the dimensionality reduction can only be achieved using *approximately* orthogonal projection matrices, while fully orthognal (left and right) would be required to not distort the outputs of the student model AND preserve cosine similarity w.rt. to the teacher.

The student is then trained to minimize cosine distance to the approximately angle preserving downprojection, on both feature and class tokens.

The method is evaluated following the approach of Proteus (which it compares to, together with Proteus V_k) , with ViT-Tiny models distilled from DINOv2 ViT-S/14 and DINOv2 ViT-B/14 teachers on ImageNet-1k, evaluated using kNN and linear probing. Ablations are performed using only Backbone, student head , a teacher head, or directly probing the teacher.

An ablation is also performed enforcing strict right-orthogonality instead of regularized approximate orthogonality, yieldig worse performance, while it is confirmed that the regularization doe syield more orthogonal matrices than unconstrained proteus.
Further fine grained clasisfication linear probin is performed on various datasets (only against proteus), consistently showing small gains. A semantic segmentation linear probing on Pascal VOC 2012 is also eprfomred, shoing improvement again.

The method also shows strong results on robustness, generalization and OOD tasks, compared with the vanilla proteus, while improving training efficiency.

Finally, the method is evaluated again (against pretrained De-IT) after being fine-tuned (2nd round f distillation after distilling with ImageNet 1k, if I understood correctly) on specialist atks (C10,C100,Food, pets), again showing strong results.

**Audience:**

Yes

**Audience Explanation:**

As models become bigger, highlighting efficiency gains that can be unlocked at all stages as in this paper is always interesting, and I think the community should appreciate the nudge towards moving beyond L2 loss towards other geometric properties .

**Broader Impact Concerns:**

I think broader impact statement would be approriate (at least for the appendix) weighing of the benefits of more efficient inference vs. jevons paradox as we see more and more energy and emissions flowing towards ML model deployment, as well as the dual use nature of segmentation and classification for surveillance via more efficient edge devices (inherent to the research of this nature, but still)

**Claims And Evidence:**

Yes

**Claims Explanation:**

The method is explained decently well, although additional details/pseudocode would be appreciated especially for the cospress finetuning setup.
I have some nits on the evaluation, but it is clear that it is effective, and while I think some related work might be missing, this is overall a good paper.

**Requested Changes:**

- multiple seeds should be run and  CIs should be reported, with suitable statistical signficicance tests performed https://arxiv.org/abs/1811.12808
- please add an algorithmic listing or pseudocode to the appendix to clarify both distillation and finetuning for readers
- I think there is some related work missing around the ideas of isomap, minimum distortion embeddings, locally linear embeddings etc. a quick search yielded the following

[1] X. He and P. Niyogi, “Locality Preserving Projections”.
[2] Y. Gao, S. Zhong, K. Hu, and J. Pan, “Robust locality preserving projections using angle-based adaptive weight method,” IET Computer Vision, vol. 14, no. 8, pp. 605–613, Dec. 2020, doi: 10.1049/iet-cvi.2019.0403.
[3] L. K. Saul, T. Labs, P. Ave, F. Park, and S. T. Roweis, “An Introduction to Locally Linear Embedding”.
[4] M. Kim and V. Pavlovic, “Covariance Operator Based Dimensionality Reduction with Extension to Semi-Supervised Settings”.
[5] J. Fischer and R. Ma, “Sailing in high-dimensional spaces: Low-dimensional embeddings through angle preservation,” Jun. 14, 2024, arXiv: arXiv:2406.09876. doi: 10.48550/arXiv.2406.09876.
[6] F. Tessari, K. Yao, and N. Hogan, “Surpassing Cosine Similarity for Multidimensional Comparisons: Dimension Insensitive Euclidean Metric,” Mar. 10, 2025, arXiv: arXiv:2407.08623. doi: 10.48550/arXiv.2407.08623.
[7] A. Agrawal, A. Ali, and S. Boyd, “Minimum-Distortion Embedding,” FNT in Machine Learning, vol. 14, no. 3, pp. 211–378, 2021, doi: 10.1561/2200000090.


I could also _swear_ that I saw specifically an inner product/angle preserving dimensionality reduction method from 2013 during the beginning of my phd, but this was in 2017 and I have not been able to find the paper (it too was trying to match the gram matrices of a kernel, similar to the approach  taken in the paper).

I think adding this context doesn't detract from the methods novelty but helps readers to put it into context

---

> ### Author Response · Authors · 2025-10-12
> **Response to Review**
>
> Thank you for your review of our paper. Please see our answers to your concerns below, and let us know if there is anything we can clarify further.
>
> > RC 1. Multiple seeds should be run and CIs should be reported.
>
> We have added experiments where multiple seeds are run, showing that variance between random seeds is small and the results are statistically significant (Table 2).
>
> > RC 2. Please add an algorithmic listing or pseudocode to the appendix.
>
> We have added a pseudocode algorithm for CosPress to the supporting information.
>
> > RC 3. Missing related work.
>
> We have added a paragraph on dimensionality reduction to the related works section that briefly covers this broader topic and some of these works.

---

### Decision · Action_Editor_fkHG · 2025-11-03

**Recommendation:** Accept as is

**Additional Comments:**

Minor typographic issues:
- Equation 2 and elsewhere, there appears to be an inconsistency between using $\cdot$ and $^\top$ for dot-products
- Equation 11, $D_{KL}$ should be $D_{\rm KL}$

**Audience:**

Yes

**Audience Explanation:**

Reviewers unanimously agreed that the paper's findings are likely of interest to the community. Feature distillation techniques have been the subject of several recent works in the vision and NLP communities. A new performant technique in this space is thus expected to be of broad interest to the community. Further, the technique has an intuitive motivation, which is interesting in and of itself, and could inspire future techniques.

**Claims And Evidence:**

Yes

**Claims Explanation:**

The paper's central claim is that when performing feature distillation for image classification, rather than matching the student embeddings with a learned projection of the teacher embeddings, it is beneficial to first learn a projection of the teacher embeddings to a lower dimensional space (via a form of SNE objective), and then to match these lower dimensional embeddings with the student embeddings. The paper presents a conceptual argument of why this approach can be superior, which is supported by empirical results.

Reviewers unanimously found the experiments to sufficiently support the main claim. Nonetheless, some limitations that were identified were:
(1) lack of clarity in exposition (e.g., precise description of algorithm)
(2) statistical significance of experimental gains being unclear
(3) experiments primarily focussed on ImageNet and ViTs (as opposed to, e.g., non self-supervised models like CNNs)

Points (1) and (2) were addressed in the revised submission, with the results now included multiple trials and more details. Point (3) was argued to reflect current practice in computer vision. We agree that results on more general architectures would be of interest; but view the current ViT results are sufficient to support the paper's claims.